# Radioembolization-Induced Changes in Hepatic [^18^F]FDG Metabolism in Non-Tumorous Liver Parenchyma

**DOI:** 10.3390/diagnostics12102518

**Published:** 2022-10-17

**Authors:** Manon N. Braat, Caren van Roekel, Marnix G. Lam, Arthur J. Braat

**Affiliations:** Department of Radiology and Nuclear Medicine, University Medical Center Utrecht, Heidelberglaan 100, 3584 CX Utrecht, The Netherlands

**Keywords:** FDG-PET, radioembolization, SIRT, PERCIST

## Abstract

Background: [^18^F]FDG-PET/CT is increasingly used for response assessments after oncologic treatment. The known response criteria for [^18^F]FDG-PET/CT use healthy liver parenchyma as the reference standard. However, the [^18^F]FDG liver metabolism results may change as a result of the given therapy. The aim of this study was to assess changes in [^18^F]FDG liver metabolism after hepatic ^90^Y resin radioembolization. Methods: [^18^F]FDG-PET/CT scans prior to radioembolization and one and three months after radioembolization (consistent with the PERCIST comparability criteria), as well as ^90^Y-PET/CT scans, were analyzed using 3 cm VOIs. The FDG activity concentration and absorbed dose were measured. A linear mixed-effects logistic regression model and logistic mixed-effects model were used to assess the correlation between the FDG-activity concentration, absorbed dose, and biochemical changes. Results: The median SUL_VOI,liver_ at baseline was 1.8 (range = 1.2–2.8). The mean change in SUL_VOI,liver_ per month with an increase in time was 0.05 (95%CI 0.02–0.09) at *p* < 0.001. The median absorbed dose per VOI was 31.3 Gy (range = 0.1–82.3 Gy). The mean percent change in ΔSUL_VOI,liver_ for every Gy increase in the absorbed dose was –0.04 (95%CI −0.22–0.14) at *p* = 0.67. The SUL_blood_ and SUL_spleen_ results showed no increase. Conclusions: The [^18^F]FDG metabolism in the normal liver parenchyma is significantly but mildly increased after radioembolization, which can interfere with its use as a threshold for therapy response.

## 1. Introduction

More and more [^18^F]FDG-PET/CT studies are being performed for response evaluations, as anatomical imaging modalities lack adequate response assessment ability or useful and consistent response criteria, or have minor prognostic value [1]. In light of these limitations of anatomical imaging, two distinct response assessment criteria were defined for [^18^F]FDG-PET: the EORTC criteria, in use since 1999 [2]; and the PERCIST criteria, in use since 2009 [1]. Their differences are outlined in Appendix A. The mean and maximum standard uptake values (SUV_mean_ and SUV_max_) corrected for body weight in a 2D region of interest (ROI) in the most metabolic part of as many tumors as possible are used in EORTC [2], while SUVs corrected for lean body mass (=SUL) in a stringent volume of interest (VOI) of 1 cm^3^ centered on the most metabolic active part of up to 5 tumors (=SUL_peak_) are used in PERCIST [1]. Additionally, PERCIST uses a SUL_mean_ assessment of the right lobe of the liver (or blood pool) for [^18^F]FDG-PET comparability at different time points and as a threshold to define the response.

In light of the PERCIST criteria and its use of a VOI in the right liver lobe to assess the comparability of [^18^F]FDG-PET, multiple test–retest studies in healthy volunteers or former cancer patients showed that the quantification of the physiological activity concentrations of [^18^F]FDG were similar over longer periods of time, including in the liver parenchyma [3]. However, studies performed in oncologic patients (mainly lymphoma patients) showed that some types of chemotherapy had an effect on the healthy liver parenchyma, leading to an increase in mean liver activity upon [^18^F]FDG-PET [4,5]. This poses a problem with the current response assessment criteria used in lymphoma patients, the Deauville criteria [6,7,8,9,10], in which the response is based on a visual assessment, comparing the tumor uptake to the liver uptake. Theoretically, the response after treatment could be wrongfully overestimated.

For the treatment of hepatic malignancies, radioembolization has gained ground in the last decades. During that time, multiple studies have used [^18^F]FDG-PET/CT for response assessments and applied the PERCIST criteria [11,12]. However, the concomitant non-target embolization and radiation of the non-tumorous liver parenchyma (NTLP) may lead to a localized and systemic inflammatory reaction [13,14]. Eventually, changes consistent with sinusoidal obstruction syndrome (SOS) and fibrosis can develop [15,16].

We hypothesized that the [^18^F]FDG activity concentration in NTLP changes after radioembolization, leading to the misinterpretation and possible overestimation of the response using this activity concentration as a threshold in the PERCIST criteria. In this study, it was investigated whether the [^18^F]FDG activity concentration in NTLP changes after radioembolization and whether alternative reference values can be used.

## 2. Methods

### 2.1. Study Population

This retrospective monocenter study comprised all patients treated with ^90^Y resin microspheres (SIRTeX Medical, Sydney, Australia) in our center from 2009 until July 2015, with an [^18^F]FDG-PET/CT prior ([^18^F]FDG-PET/CT_prior_) and one or three months after treatment ([^18^F]FDG-PET/CT_follow-up_). Patients were only included if the [^18^F]FDG-PET/CT scans met the PERCIST criteria (Appendix A) and if a ^90^Y-PET/CT was performed within 24 h of radioembolization for a regional dose calculation of the NTLP.

The institutional medical ethics committee waived the need for informed consent for this retrospective review.

### 2.2. Radioembolization Treatment

After a pre-treatment hepatic arteriography to evaluate the vascular anatomy, a simulation was performed with the injection of ^99m^Tc-macroaggregated albumin (MAA) into the hepatic artery or arteries. Standard planar and SPECT images of ^99m^Tc-MAA were obtained for particle distribution assessments and to exclude relevant liver–lung shunting and extrahepatic depositions. In a separate session, ^90^Y resin microspheres were injected into the hepatic artery or arteries after ensuring adequate catheter placement (similar to the ^99m^Tc-MAA injection position). The therapeutic activity of ^90^Y resin microspheres was calculated according to the body surface area method.

### 2.3. [^18^F]FDG-PET/CT Imaging Protocols

All [^18^F]FDG-PET/CT scans at all time points were performed on the same PET/CT-scanner (Biograph mCT, Siemens Healthcare, Erlangen, Germany). All patients fasted for at least 6 h prior to intravenous [^18^F]FDG administration (2.0 MBq/kg) and each patient’s blood glucose level was determined prior to the tracer injection (<11.1 mmol/L). The imaging parameters included a three-dimensional acquisition technique with a 216 mm field of view, 3 min per bed position, and ordered subset expectation maximization iterative reconstruction, including a Gaussian filter, 4 iterations, and 21 subsets. The measurements were performed on image reconstructions according to the EARL criteria [17].

### 2.4. ^90^Y-PET/CT Imaging Protocol

All ^90^Y-PET/CT’s were performed on the same PET/CT-scanner (Biograph mCT, Siemens Healthcare, Erlangen, Germany) within 24 h of treatment. The imaging parameters included a total acquisition time of 30 min for 2 bed positions, TrueX and time-of-flight reconstructions, and a reconstruction using 4 iterations with 21 subsets and a 5 mm full-with at half maximum Gaussian post-reconstruction filter. A low-dose CT scan was acquired for attenuation correction and as an anatomical reference.

### 2.5. Healthy Liver Parenchyma Analysis

All [^18^F]FDG-PET/CT and ^90^Y-PET/CT scans were analyzed using ROVER software (ABX, Radeberg, Germany). A 3 cm VOI in the right lobe of a normal liver is a robust indicator for the identification of the SUL_mean_ [18] and is applied in the PERCIST criteria as a threshold parameter. To assess the NTLP activity concentration, the liver was separated into three regions, consistent with the three supplying arteries: the left hemiliver (i.e., the left hepatic artery (LHA)), the right hemiliver (i.e., the right hepatic artery (RHA)), and segment 4 (i.e., the middle hepatic artery (MHA) or segment 4 artery). Segment 4 was considered a separate region, as the MHA can be separately injected.

Spheric VOIs (3 cm in diameter) were placed in the NTLP in all three liver regions and in the spleen. If one of the regions was too diffusely involved in malignancy to accommodate a 3 cm VOI, no VOI was placed. In cases of (extended) right-sided hemihepatectomy, a second VOI was placed in the remaining liver tissue. Cilindrical VOIs with a 2 cm diameter were placed in the ascending, descending, and abdominal aortas, without inclusion of the aortic wall (to avoid elevated [^18^F]FDG-uptake due to atherosclerosis).

The [^18^F]FDG-PET/CT_follow-up_ of each patient was registered to the [^18^F]FDG-PET/CT_prior_. ROIs were automatically co-registered onto the [^18^F]FDG-PET/CT_follow-up_, but were manually corrected if the automatic placement of the VOI did not correspond to the location of the VOI on the [^18^F]FDG-PET/CT_prior_.

The SUL_mean_ and SUL_max_ were calculated for all VOIs.

The same parameters were registered for the spleen, as well as the splenic volume. The splenic volume is known to increase after radioembolization; however, it is unknown whether the [^18^F]FDG activity concentration changes are correlated to splenic volume changes [19].

In addition, all ^90^Y-PET/CT scans were registered to the [^18^F]FDG-PET/CT_prior_ and the liver VOIs were co-registered onto the ^90^Y-PET/CT, enabling a read-out of the absorbed dose (MBq/mL) in the same three regional VOIs. In a previous report, we validated the use of ROVER software for absorbed dose estimation on ^90^Y-PET/CT [20].

All measurements on the [^18^F]FDG-PET/CT_prior_ were performed by one physician (MB; >10 years of experience).

### 2.6. Biochemical Changes

Laboratory examinations at the time of each PET/CT were noted, including liver function tests (total bilirubin, alkaline phosphatase, gamma-glutamyl transferase (GGT), aspartate aminotransferase (AST), alanine aminotransferase (ALT)) and coagulation parameters (thrombocytes, international normalized ratio (INR), partial thromboplastin time (PTT), activated partial thromboplastin time (APTT), and thrombin time (TT)). The Common Terminology Criteria for Adverse Events (CTCAE) version 5.0 was used to grade biochemical toxicities. The presence of ≥2 CTCAE grade ≥1 for laboratory adverse events was regarded as clinically relevant biochemical toxicity.

Biochemical changes were correlated with changes in [^18^F]FDG uptake in the NTLP.

### 2.7. Statistical Analysis

Descriptive statistics were used to explore baseline and treatment characteristics. The changes in SUL_liver_ (the average of all VOIs per patient), SUL_VOI,liver_, SUL_spleen_, and SUL_blood_ over time were assessed using linear mixed-effects regression models to account for clustered data. Nested models were compared using Akaike’s Information Criterion. A model with a random slope only fitted the data best for all parameters. Time was used as a continuous independent variable. In the analyses of SUL_liver_ and SUL_VOI,liver_, the injected activity (MBq) (patient-level) and the absorbed dose per VOI (VOI-level) were included as co-variables to adjust for possible confounding.

The association between the parenchymal-absorbed dose and the change in SUL_VOI,liver_ (represented as ΔSUL_VOI,liver_) was also analyzed using a linear mixed-effects regression model with a random slope, with the absorbed dose as the continuous independent variable.

A logistic mixed-effects model was used to assess the association between the presence of clinically relevant biochemical toxicity (using the cut-off definition as described above) per time point and the change in SUL_VOI,liver_. The analysis was adjusted for response to therapy (coded as yes/no for progressive disease) as a possible confounder. All statistical analyses were done with R statistical software version 3.6.2 for Windows. We report here the effect estimates with associated 95% confidence intervals and corresponding two-sided *p*-values.

## 3. Results

A total of 292 patients were screened, of whom 43 patients underwent an [^18^F]FDG-PET/CT_prior_ and ≥1 [^18^F]FDG-PET/CT_follow-up_ and an ^90^Y-PET/CT. Sixteen patients were excluded because they did not meet the PERCIST criteria for inter-study comparisons of [^18^F]FDG-PET/CT. One additional patient was excluded for having liver metastases that were too diffuse to accommodate a 3 cm VOI in the NTLP. Twenty-six patients were included, with 35 [^18^F]FDG-PET/CT_follow-up_ scans (a total of 61 [^18^F]FDG-PET/CTs), resulting in a total of 62 VOI_liver_ scans that were analyzed in this study (Figure 1). Nine patients had two [^18^F]FDG-PET/ CT_follow up_ scans, while the remaining 17 patients had one [^18^F]FDG-PET/ CT_follow up_ scans at 1 month (*n* = 9) or 3 months (*n* = 8). The baseline characteristics are presented in Table 1.

The median post-injection times were 64.5 min, 62.3 min, and 65 min at [^18^F]FDG-PET/CT_prior_ and the 1 and 3 month [^18^F]FDG-PET/CT_follow-up_ scans, respectively (Table 2). The median absorbed dose per VOI was 31.3 Gy (range = 0.1–82.3 Gy). Overall, the median SUL_VOI,liver_ at baseline was 1.8 (range = 1.2–2.8). A significant increase in SUL_VOI,liver_ over time was found; the mean change in SUL_VOI,liver_ per month increase in time was 0.05 (95%CI 0.02–0.09) at *p* = 0.00088 (Figure 2)(Table 3). The mean percent change in ΔSUL_VOI,liver_ for every Gy increase in dose was −0.04 (95%CI -0.22–0.14) at *p* = 0.67. At the patient level, the median SUL_VOI_ (average of all VOIs of the liver parenchyma) was 1.8 (range = 1.5–2.3). The mean change in SUL_VOI_ per month increase in time was 0.05 (95%CI 0.01–0.08); *p* = 0.016 (Table 3).

Overall, the incidence of biochemical toxicity was low. CTCAE grade 3 toxicity was only found for bilirubin, GGT, and APTT (Table 4). PTT, INR, and TT were slightly prolonged in three, three, and one patient(s), respectively. The results of the logistic regression model showed a non-significant association between the clinically relevant biochemical toxicity and change in SUL_VOI_ (at the patient level) (95%CI 0.0006–333.29) at *p* = 0.90 (Table 5) (Figure 3).

For SUL_blood_, no significant increase or decrease over time was found; the mean change in SUL_blood_ per month increase in time was −0.012 (95%CI −0.05–0.03) at *p* = 0.57 (Figure 4A). SUL_spleen_ was also stable over time; the mean change per month increase over time was 0.005 (95%CI −0.04–0.05) at *p* = 0.83 (Figure 4B), while the splenic volume did not significantly increase over time. There was a 49 mL increase in volume per month (95%CI 32–65, *p* = 0.0011) (Figure 4C).

## 4. Discussion

Our data showed that after ^90^Y radioembolization, the [^18^F]FDG activity concentration (SUL_VOI liver_) in the collateral targeted NTLP was significantly increased compared to baseline. Although the SUL_VOI liver_ increase was minimal, with a mean change in SUL_VOI,liver_ per month increase over time of 0.05 with a median SUL_VOI,liver_ at baseline of 1.8, this may influence quantitative assessments of liver activity concentrations as used in PERCIST. The influence on routine visual response assessments remains unclear, but will be minimal. Mild changes in [^18^F]FDG activity concentration, however, could be noticed in our study visually (Figure 5).

Multiple reports on increased [^18^F]FDG uptake in the liver after chemoradiation therapy for esophageal cancer have been published, with focal as well as diffuse patterns [16,21,22,23,24]. However, only one publication has touched upon the [^18^F]FDG uptake pattern in normal liver parenchyma 1 month after radioembolization, with no changes compared to baseline [25]. Unfortunately, additional data were lacking and no correlation was made with the absorbed dose.

Remarkably, we did not find an association between the median absorbed dose and the change in SUL_VOI,liver_. In contrast, Nakahara et al. reported a direct visual correspondence to the radiation dose distribution and the pattern of increased [^18^F]FDG uptake in the liver in a patient after chemoradiation therapy for esophageal cancer [24]. The lack of association in our data may have been due to the limited patient numbers and the limited spread in absorbed doses between the VOIs.

The explanation for the increased [^18^F]FDG liver activity concentration after radioembolization is largely unknown. Local inflammation seems plausible, although this is not confirmed in the existing (although scarce) histopathology literature [15,26,27,28]. An in vitro assessment of ovarian cancer cell lines after radiotherapy showed swelling and increased [^3^H]FDG uptake in the surviving cells [29]; a similar phenomenon may be present in hepatocytes after radioembolization. However, the most plausible explanation may be increased [^18^F]FDG uptake due to sinusoidal obstruction syndrome (SOS). Several authors reported signs of SOS developing within the first 3 months after radioembolization [15,27,28]. As shown by Kim et al. in 35 patients with SOS after platin-based chemotherapy, SOS can lead to a significantly increased [^18^F]FDG liver activity concentration [30]. The possible explanation they offer is passive [^18^F]FDG tracer stasis due to endothelial cell injury and peliotic changes. The increase in SUL_liver_ in their study (12%) was considerably higher than in our data, yet with similar to our findings; the activity concentration in the blood pool remained unchanged, unlike the splenic activity concentration (which increased significantly in moderate to severe cases of SOS in their study). As the blood pool uptake seems unchanged at different time points after injection, in different patient populations and even after chemotherapy, it may be the best reference value for patients treated with radioembolization [4,6,7,8,9,30,31].

Although PERCIST aims to unify response assessments with [^18^F]FDG-PET, in our study 30.8% of the patients with [^18^F]FDG-PET/CT scans at baseline and follow-up were excluded based on the PERCIST timing and scan criteria (apart from the liver SUL criterion). This is consistent with most publications, which show mean exclusion percentages of 22.6–38.3% for PERCIST response assessments, due to the strict criteria for scan comparability (Appendix A) [9,10,32]. The major restrictions include the difference in post-injection scanning time (<15 min) and the minimum of 50 min needed for post-injection scanning. Unfortunately, these are significant factors in SUV measurements and major logistical problems for most nuclear medicine departments. As shown in several test–retest studies, the interval between injection and acquisition is a significant parameter for underestimating liver activity [31,32,33,34]. This may be explained by the abundance of the enzyme glucose-6-phoshphatase in the liver, causing continuous glycolysis and a decrease in FDG retention [35]. Thus, it is essential to minimalize the differences in FDG uptake time between scans to ensure a consistent threshold for PET-based response assessments (as used in PERCIST, the Deauville criteria, etc.). The post-injection times between [^18^F]FDG-PET/CT scans in our study were not significantly different (Table 2). However, in 5/26 of the right liver VOIs, the SUL_VOI, liver_ showed a ≥0.3 SUL unit change, violating the scan restrictions according to the PERCIST criteria. In these cases, the radioembolization-induced increased [^18^F]FDG liver activity concentration would have resulted in exclusion. However, these minimal changes in SUL_VOI_ will probably not be clinically relevant in routine visual response assessments. With the increased use of the modified PERCIST criteria, the reference threshold of the VOI_liver_ is lowered, and the use of a more robust reference value in clinical studies is advisable [36].

A comparison of [^18^F]FDG liver activity concentrations with earlier reported results is difficult, due to the differences in the quantification or correction of the [^18^F]FDG uptake, i.e., the use of ROIs or VOIs; the use of SUV_max_, SUV_mean_, or the total lesion/liver glycolysis (TLG); and correction for the body surface area (BSA) vs. body weight vs. lean body mass (LBM). Only the LBM-corrected SUV (SUL) is not significantly affected by body weight, contrary to the BSA-corrected or body-weight-corrected SUV. The calculations are not affected by age, blood glucose, diabetes, or gender [3,4,31]. Therefore, in our study, SUL was chosen for all measurements, consistent with the PERCIST criteria.

There were a few limitations to this study, the foremost of which was the small patient number with consistently acquired PET/CT scans according to the PERCIST guidelines. A patient cohort with only lobar treatments, enabling an in-patient comparison with non-treated NTLP, would be better by avoiding inter-patient biological differences. Additionally, a comparison of patients treated with different commercially available microspheres (with their differing specific activity levels) could help to further evaluate the role of [^18^F]FDG PET/CT as a follow-up tool after radioembolization. Given the expanding use of both external beam radiotherapy and radioembolization and the increasing use of PET/CT for response assessments, further research is required.

## 5. Conclusions

The [^18^F]FDG liver activity concentration in the background liver (SUL_VOI_ ) is mildly but significantly increased after radioembolization; therefore, this can interfere with the use of the background liver results firstly as a threshold for therapy response and secondly as a comparability criterion in the PERCIST criteria. An alternative and more consistent threshold is the [^18^F]FDG activity concentration in the blood pool (SUL_blood_).

## Figures and Tables

**Figure 1 diagnostics-12-02518-f001:**
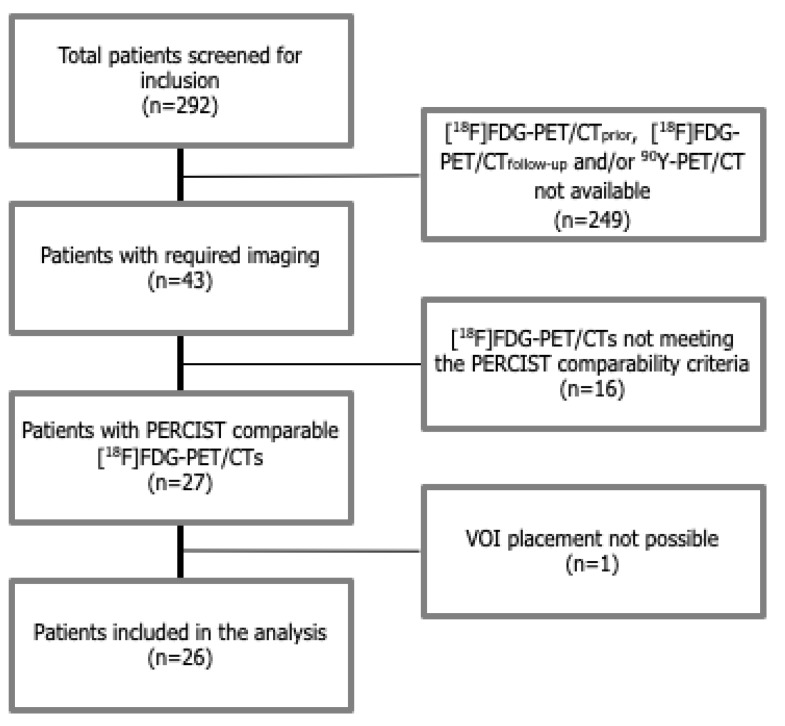
Flowchart of the study.

**Figure 2 diagnostics-12-02518-f002:**
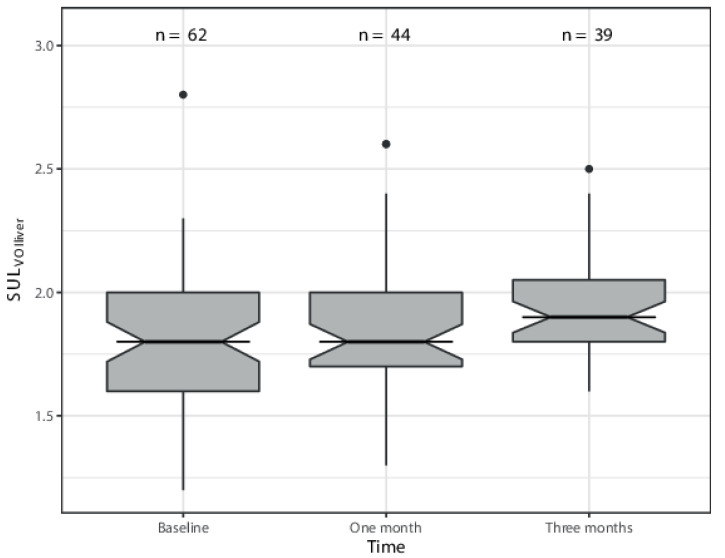
Changes in SUL_VOI,liver_ over time compared to baseline values.

**Figure 3 diagnostics-12-02518-f003:**
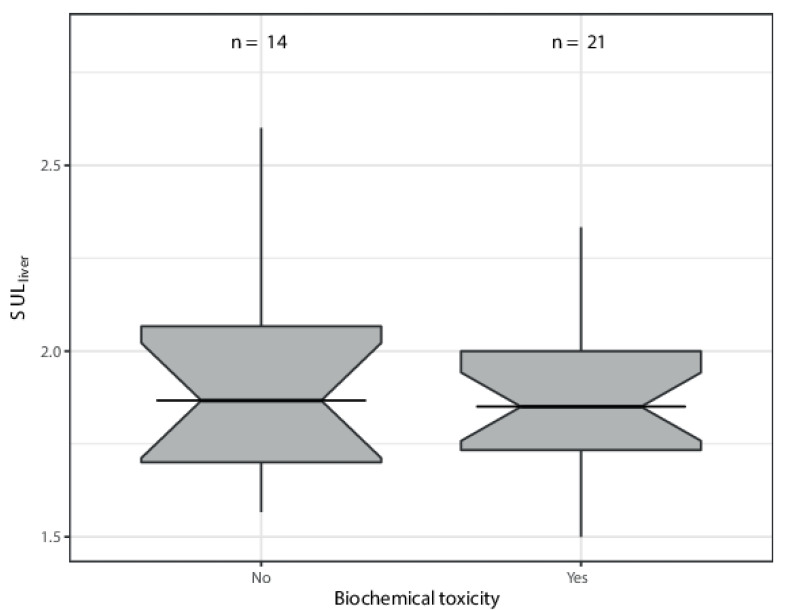
Differences in SUL_liver_ (patient-level) versus biochemical toxicity. Numbers represent the different durations of biochemical toxicity determination per patient. Some patients had measurements at two follow-up times so numbers do not equal total number of patients.

**Figure 4 diagnostics-12-02518-f004:**
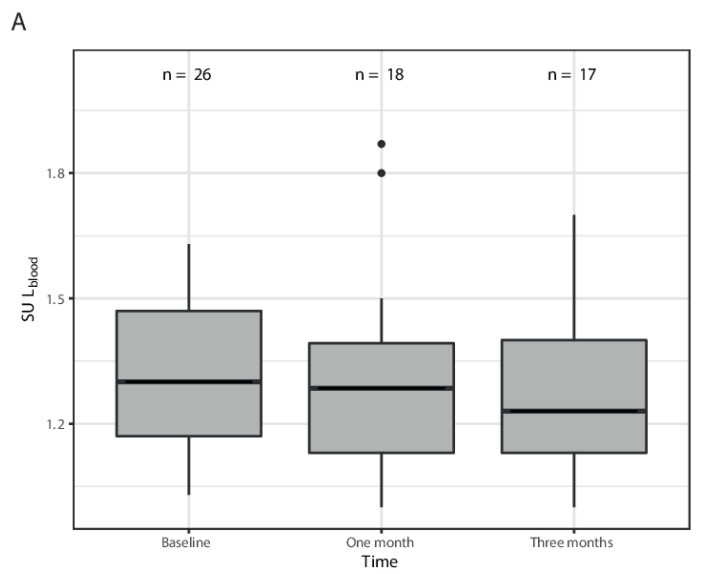
Changes in SUL_blood_ (**A**) and SUL_spleen_ (**B**) over time compared to baseline and changes in splenic volume (**C**) over time compared to baseline.

**Figure 5 diagnostics-12-02518-f005:**
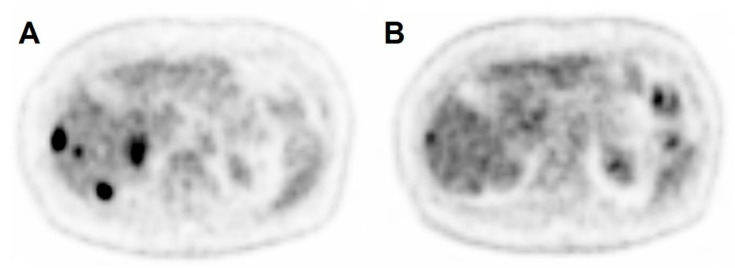
Case example of visual increase in [^18^F]FDG activity concentration in non-tumorous liver tissue. A 75 year old woman with colorectal liver metastases underwent a [^18^F]FDG-PET/CT_prior_ (**A**) and a [^18^F]FDG-PET/CT_follow-up_ (**B**) 15 days prior and 29 days after radioembolization, respectively. The images are similarly scaled (0–7 SUV_lean body mass_). One month after radioembolization, a reduced [^18^F]FDG activity concentration can be seen in the metastases, but the background non-tumorous liver activity concentration has visually increased (SUL_VOI,liver_ increase from 1.8 to 2.4 in the VOI_right liver_).

**Table 1 diagnostics-12-02518-t001:** Baseline characteristics of the included patients.

Total Number of Evaluable Patients	26
Male/Female	16/10
Median age in years (range)	66 (34–79)
Median time from baseline F-18 FDG-PET/CT to Y-90-RE in days (range)	24 (1–44)
Median weight in kg (range)	82 (57–110)
Number of patients known with diabetes mellitus	1
Number of patients with use of anti-diabetic medication	1
Primary tumor - Colorectal carcinoma - Pancreatic adenocarcinoma	25 1
Prior locoregional treatment - Extended right hemihepatectomy - Left hemihepatectomy + one metastasectomy in the right hemiliver - Multiple metastasectomies - Radiofrequency ablation	3 1 3 1
Median injected activity in MBq (range)	1484 (345–2164)
Treatment - Whole liver delivery - Sequential delivery - Lobar treatment	22 1 3
Prior systemic treatment	26
No. of systemic treatment lines - 1 - 2 - 3 - 4	9 12 4 1

Legend: RE = radioembolization.

**Table 2 diagnostics-12-02518-t002:** F-18 FDG-PET/CT specifications and measurements of the study population.

Time After RE	Baseline	One Month	Three Months
Number of examinations	26	18	17
	*Median (range)*		
Serum glucose level in mmol/L	6.4 (5.1–9.8)	6.3 (4.2–10.3)	6.6 (4.4–10.4)
Post-injection time in minutes	64.5 (56–85)	62.5 (58–78)	65 (51–86)
Injected activity in MBq/kg	2.03 (1.81–2.37)	2.03 (1.66–2.50)	1.94 (1.69–2.62)
SUL_mean_ segment 2-3	1.8 (1.5–2.2)	1.9 (1.5–2.6)	1.95 (1.6–2.4)
SUL_mean_ segment 4	1.7 (1.5–2.1)	1.7 (1.6–2.1)	1.8 (1.6–2.2)
SUL_mean_ right hemiliver	1.9 (1.2–2.8)	1.7 (1.3–2.6)	1.9 (1.7–2.5)
SUL_mean_ spleen	1.4 (1.1–2.0)	1.5 (0.9–1.9)	1.5 (1.3–1.9)
SUL_mean_ bloodpool	1.3 (1.03–1.63)	1.27 (1.00–1.87)	1.23 (1.00–1.70)

**Table 3 diagnostics-12-02518-t003:** Relation of the SUL_VOI_ over time based on linear mixed effects regression analyses.

Independent Variable	Mean Change in SUL_VOI_ Per Month Increase in Time (95%CI); *p*-Value	
SUL_VOI_ (patient-level)	*Unadjusted*	*Adjusted (for administered activity Y90)*
	0.05 (0.01–0.08); 0.016	0.03 (−0.04–0.1); 0.38
SUL_VOI,liver_ (VOI-level)	*Unadjusted*	*Adjusted (for absorbed dose per VOI)*
	0.05 (0.02–0.09), 0.00088	0.05 (0.02–0.08); 0.00084

Legend: Numbers represent the different durations of biochemical toxicity determination per patient. Some patients had measurements at two follow-up times, so numbers do not equal total number of patients.

**Table 4 diagnostics-12-02518-t004:** Incidence rates of biochemical toxicity between baseline and three months after radioembolization according to CTCAE version 5.

CTCAE Grade	1	2	3
Alanine aminotransferase	11		
Alkaline phosphatase	6	3	
Aspartate aminotransferase	14	1	
Bilirubin	5	1	1
Gamma-glutamyl transferase	3	5	3
Platelets	9	2	
APTT	13	1	1

**Table 5 diagnostics-12-02518-t005:** Relation between SUL_VOI,liver_ and clinically relevant biochemical toxicity based on mixed-effects logistic regression analyses with SUL_VOI,liver_ as the independent variable.

	*Clinically Relevant Biochemical Toxicity Yes Versus No*	
**Dependent variable**	**Odds ratio for clinically relevant biochemical toxicity for every month increase in SUL_VOI,liver_ (95% CI); *p*-value**	
	*Unadjusted model*	*Adjusted model (for progressive disease)*
Clinically relevant biochemical toxicity (based on laboratory parameters) (no *n* = 14, yes *n* = 21)	0.42 (0.0006–296.82); 0.69	0.48 (0.0006–333.29); 0.90

## Data Availability

The data presented in this study are available on reasonable request from the corresponding author.

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
