# Peer review of "Radioembolization-Induced Changes in Hepatic [18F]FDG Metabolism in Non-Tumorous Liver Parenchyma"

_diagnostics, 2022, doi:10.3390/diagnostics12102518_

Round 1

Reviewer 1 Report

I think that your work is of much interest considering that PERCIST is strongly based on comparison to normal tissue uptake.

As you mentioned there are many factors influencing the NTLP uptake, including other local or systemic therapies so there is much work to do on the topic.

Even more recent references are lack on this area.

I will make minor changes to the conclusion.

Instead of

18F]FDG metabolism in the normal liver parenchyma is significantly, but mildly increased after radioembolization, and therefore can interfere with its use as a threshold for therapy-response.

I will rather say

18F]FDG metabolism in the normal liver parenchyma is mildly but significantly increased after radioembolization, and therefore can interfere with its use as a threshold for therapy-response.

Reviewer 2 Report

The authors wrote an useful manuscript about an interesting topic. The results won't change the world but they are essential to the scientific community in a way that they confirm that the threshold can be used for therapy-response but that this should be done with caution.

Minor things;

- M&M line 100 18F-FDG imaging protocols the details as given for the 90Y protocol are missing here.

- how do the authors know that the small alterations found are not due to changes in blood glucose level (as stated in line 291 but how do the authors know for sure?) or injected activity? Or more analysis related, due to the placement of the VOI. It is not clear if the authors investigate the impact of intra-observer variation.

- the question remain whether statistical significant is in this regard also clinical relevant. The authors could eleborate a bit more on this in the discussion.

- The last sentence of the conclusion states the option of using the SULblood instead of the SUL voi-liver. This is not discussed in the discussion section and should be.

- only 4/35 literature references are published <5y ago more recent literature should be added.

Author Response

Thank you for your comment. As you proposed, we altered the first sentence of the conclusion to:

[18F]FDG liver activity concentration in the background liver (SULVOI ) is mildly, but significantly increased after radioembolization, and therefore can interfere with the use of the background liver firstly as a threshold for therapy-response and secondly as a comparability criterion in the PERCIST criteria.

Round 2

Reviewer 2 Report

-